# Counterfactually Augmented Event Matching for De-biased Temporal Sentence Grounding

Xun Jiang
Center for Future Media & School of
Computer Science and Engineering
University of Electronic Science and
Technology of China
Chengdu, China

Zhuoyuan Wei
Center for Future Media & School of
Computer Science and Engineering
University of Electronic Science and
Technology of China
Chengdu, China

Shenshen Li
Center for Future Media & School of
Computer Science and Engineering
University of Electronic Science and
Technology of China
Chengdu, China

Xing Xu*
Center for Future Media & School of
Computer Science and Engineering
University of Electronic Science and
Technology of China
Chengdu, China
College of Electronic and Information
Engineering
Tongji University
Shanghai, China

Jingkuan Song
Center for Future Media & School of
Computer Science and Engineering
University of Electronic Science and
Technology of China
Chengdu, China
College of Electronic and Information
Engineering
Tongji University
Shanghai, China

Heng Tao Shen
Center for Future Media & School of
Computer Science and Engineering
University of Electronic Science and
Technology of China
Chengdu, China
College of Electronic and Information
Engineering
Tongji University
Shanghai, China

## Abstract

Temporal Sentence Grounding (TSG), which aims to localize events in untrimmed videos with a given language query, has been widely studied in the last decades. However, recently researchers have demonstrated that previous approaches are severely limited in out-of-distribution generalization, thus proposing the *De-biased TSG* challenge which requires models to overcome weakness towards outlier test samples. In this paper, we design a novel framework, termed **C**ounterfactually **A**ugmented **E**vent **M**atching (**CAEM**), which incorporates counterfactual data augmentation to learn event-query joint representations to resist the training bias. Specifically, it consists of three components: (1) A Temporal Counterfactual Augmentation module that generates counterfactual video-text pairs by temporally delaying events in the untrimmed video, enhancing the model's capacity for counterfactual thinking. (2) An Event-Query Matching model that is used to learn joint representations and predict corresponding matching scores for each event candidate. (3) A Counterfact-Adaptive Framework (CAF) that incorporates the counterfactual consistency rules on the matching process of the same event-query pairs, furtherly mitigating the bias learned from training sets. Extensive experimental results conducted on two widely used DTSG datasets, *i.e.*, Charades-CD and ActivityNet-CD, show that our our proposed CAEM method outperforms recent

state-of-the-art methods. Our implementation code is available at https://github.com/CFM-MSG/CAEM_Code.

## CCS Concepts

• **Computing methodologies** → **Activity recognition and understanding**; *Causal reasoning and diagnostics.*

## Keywords

De-biased Temporal Sentence Grounding; Counterfactual Reasoning; Multimodal Learning; Video Understanding;

**ACM Reference Format:**
Xun Jiang, Zhuoyuan Wei, Shenshen Li, Xing Xu, Jingkuan Song, and Heng Tao Shen. 2024. Counterfactually Augmented Event Matching for De-biased Temporal Sentence Grounding. In *Proceedings of the 32nd ACM International Conference on Multimedia (MM '24), October 28-November 1, 2024, Melbourne, VIC, Australia.* ACM, New York, NY, USA, 10 pages. https://doi.org/10.1145/3664647.3680948

## 1 Introduction

Temporal Sentence Grounding (TSG) task [1–4] aims at localizing video events from untrimmed long videos in terms of given text queries. In previous works, the TSG models followed a prior assumption that the training and testing samples are under Independent-Identical-Distribution (IID) settings. However, test samples from the real-world environment are unpredictable, which could be outlier samples and lead to Out-Of-Distribution (OOD) problems. Recently, researchers [5, 6] demonstrated that the conventional TSG methods suffer from heavily biased training data and show limited generalization on outlier data. As is illustrated in Fig. 1(a), the target moments in IID data share a similar statistical distribution with the training data, while the target moments in OOD data show a shifted distribution, simulating the open-world outlier test samples. According to the initial works [5, 6] on exploring the bias in TSG

---

*Corresponding Author.

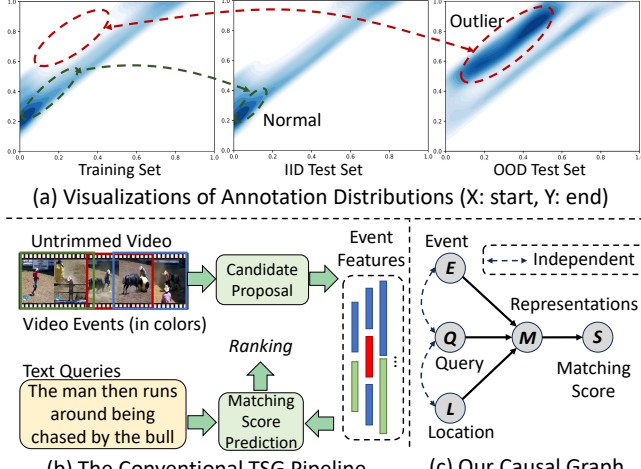

(a) Visualizations of Annotation Distributions (X: start, Y: end)

(b) The Conventional TSG Pipeline

(c) Our Causal Graph

**Figure 1: (a) An illustration of the Training, Test-IID, and Test-OOD distributions of the DTSG challenge. (b) The conventional TSG pipeline first generates event candidates and then predicts matching scores. (c) Our causal graph, where $E, Q, L, M$, and $S$ represent Events, Queries, Locations, Multimodal Representations, and Scores respectively.**

task, the previous TSG methods [1, 3, 4, 7, 8] heavily overfit spurious correlations between video-text pairs and manual temporal annotations, leading to dramatically dropped performance on OOD test samples. To this end, exploring De-biased TSG (DTSG) methods, which could overcome the weakness of model generalization, is becoming an emerging task, raising wide attraction to the community of multimodal learning and multimedia applications.

Following the mentioned initial works [5, 6], a group of works [5, 6, 9–11] are proposed to tackle the DTSG task. Most of these methods incorporated causal reasoning [5, 6, 12] or data augmentation [9, 10] into the TSG task and achieved promising de-biased ability in the challenging DTSG task. Although these methods have made promising progress in avoiding the direct influence of annotation bias on prediction results, they ignore the indirect impact of the local location of visual content on the learned event-query joint representations, which is an essential intermediate process for conventional TSG paradigm. As illustrated in Fig. 1(b), the widely adopted TSG paradigm, *i.e.*, proposal-based pipeline [1, 3, 4, 7], firstly generates video candidates and then assigns the matching score for ranking. In such a paradigm, the TSG models are trained to discriminate well-matched event-query pairs, instead of predicting the temporal location directly. Hence, we consider the causal graph in Fig. 1(c), where the joint representations have three independent parent nodes: text query, video event and event location.

In this paper, we propose a novel DTSG approach for resisting training bias in learning event-query matching, termed Counterfactually Augmented Event Matching (CAEM). Concretely, we first devise a *Temporal Counterfact Augmentation (TCA)* module to facilitate TSG models with counterfactual thinking ability. We observed that the process of annotating moments, which is actually matching event-query pairs manually, depends on the currently observed video content sequence. However, such a process forgo teaching the model the ability of counterfactual thinking, i.e., *What*

*if the target moment does not happen at this time but in the rest of the given videos?* To this end, we propose the TCA module to generate counterfactuals by delaying the observed events to later moments. The counterfactuals are used to train our models with the observed factual data jointly. Secondly, we design an *Event-Query Matching (EQM)* model that follows the proposal-based TSG pipeline illustrated in Fig. 1(b) to predict event-query matching scores. Finally, considering the causal graph depicted in Fig. 1(c), we introduce the *Counterfactual Adaptive Framework (CAF)*, which learns event-query joint representations and integrates counterfactual consistency rules [13, 14] to ensure consistent semantics for identical events occurring in different contexts.

We evaluate our proposed CAEM method on two widely used DTSG benchmark datasets, *i.e.*, Charades-CD and ActivityNet-CD. Additionally, we also generalize it on two benchmark datasets with novel text queries. Extensive experimental results prove our method outperforms recent state-of-the-art methods on the DTSG task.

Overall, our contributions can be summarized as follows:

- We propose a novel method termed Counterfactually Augmented Event Matching (CAEM) for the De-biased Temporal Sentence Grounding task. It is significantly bias-resist on imbalanced training data and remarkably improves the generalization ability toward OOD test samples.
- We devise a Temporal Counterfact Augmentation that creates counterfactual perturbations on temporal locations of video events. It effectively introduces counterfactual thinking in TSG models thus improving generalization ability.
- We design a Counterfact-Adaptive Framework that learns event-query joint representations in both observed and counterfactual training samples. It follows the counterfactual consistency rule to maintain semantical consistency.

## 2 Related Works

**Out-Of-Distribution Generalization.** The OOD generalization endeavors to train a model using data from the training environments so that it can effectively generalize to unfamiliar environments. As a challenging but practical problem, the OOD generalization, numerous algorithms [15–17] have been devised to enhance OOD generalization. One set of methodologies concentrates on reducing the disparities among the training environments [15, 18]. The meta-learning domain generalization method [19] utilizes a meta-learning approach and introduces simulated training and testing distributional shifts during training. In [16, 17, 20], robust optimization is employed to train models for minimizing the worst-case training loss across a predetermined set of groups, while several approaches [18, 21] incorporate adversarial training to enhance OOD generalization performance. Additionally, the OOD generalization has been also widely explored in multimodal learning tasks, such as multimodal fusion analysis [22–25], vision-language understanding [26–28], or multimedia content retrieval [5, 6, 29–31], *etc.*. Particularly, Yuan *et al.* [5] and Yang *et al.* [6] raised a discussion on the temporal out-of-distribution problems in the temporal sentence grounding task, which motivated us to develop this work.

**De-biased Temporal Sentence Grounding.** As a fundamental problem in multimodal video understanding, the temporal sentence grounding task [3, 4, 32, 33] has been studied for several years

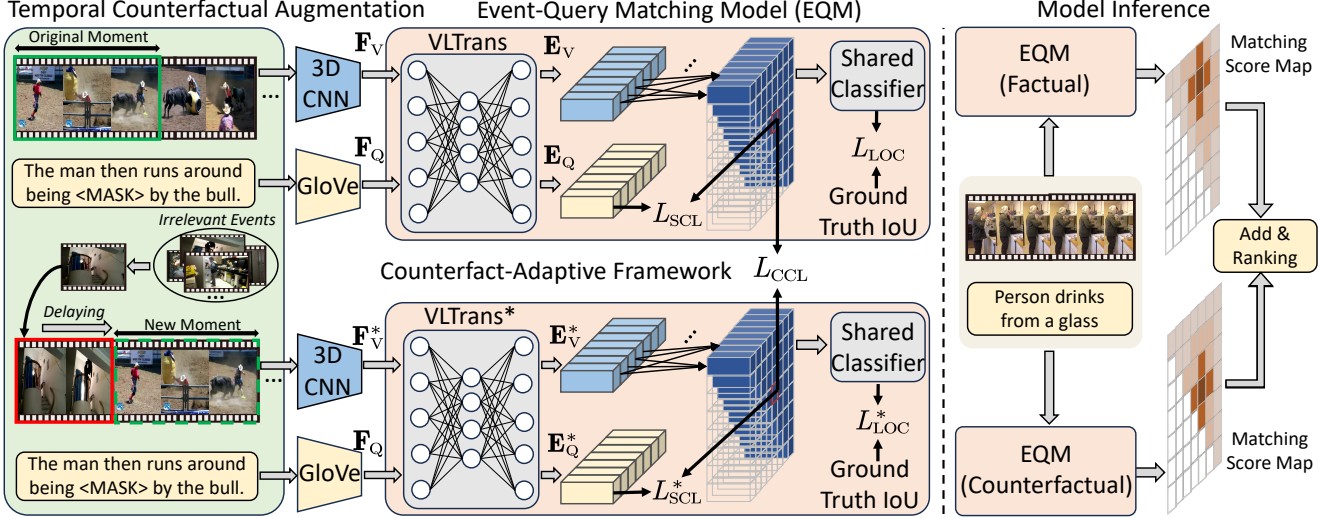

**Figure 2: An illustration of our proposed CAEM method. It consists of three key components: (1) Temporal Counterfactual Augmentation which aims to generate counterfactuals where the target events are delaying, (2) Event-Query Matching model which is used for learning event-query joint representations and predicting matching scores, and (3) Counterfact-Adaptive Framework that maintains consistent semantics of identical events occurring in different moments.**

and achieves promising performance via manual annotations with sufficient temporal cues. However, recent studies [5, 6] empirically demonstrated that the previous methods have defects in outlier test samples, showing limited generalization on unpredictable samples from the real world. To this end, more researchers focus on a more challenging task, *i.e.*, De-biased Temporal Sentence Grounding (DTSG) [9–11, 34], which aims at resisting training bias and improving model generalization on outlier test data. Specifically, Yuan *et al.* [5] proposed the initial DTSG work on exploring the performance of TSG models on IID and OOD test samples and proposed a corresponding solution for this challenge. Yang *et al.* [6] proposed a causality-inspired TSG framework that builds a structural causal model to capture the true effect of query and video content on the prediction. Nevertheless, all these DTSG methods studied the direct impacts on temporal predictions caused by biased training data, but ignore the truth that TSG methods select predictions by evaluating the matching scores of event-query joint representations. Inspired by this observation, we aim at developing a more effective DTSG method against training bias.

**Counterfactual Data Augmentation.** The concept of counterfactuals originates from causal inference [35–37], where it pertains to hypothetical situations contrary to observed facts. In the context of data augmentation, counterfactuals represent plausible variations of existing data points that maintain their fundamental characteristics while introducing changes that are consistent with the underlying data distribution. In recent years, CDA has garnered significant attention across various domains, including recommendation systems [38–40], natural language processing [41–43], and multimodal learning [44–47]. For example, Wang *et al.* [38] proposed a CDA-based framework to mitigate the impact of imperfect training data and empower sequential recommendation models. Dixit *et al.* [41] devised a two-stage counterfactual data augmentation for large-scale pre-trained language models, achieving better generalization to OOD data. Overall, by exploring counterfactual

spaces, CDA enables models to learn from a more diverse range of scenarios, thereby improving model robustness to handle unseen instances and adapt to real-world outlier data. To this end, we incorporate the CDA into DTSG task for improving the resistency against training bias caused by manual annotations.

## 3 Proposal Method

### 3.1 Preliminary

**Problem Formulation.** We represent the untrimmed video set as $V$, while the text query set is represented as $Q$. For each case, the input can be represented as a pair consisting of untrimmed video $v$ and a sentence query $q$. Let $t_s$ and $t_e$ be the start and end time of one target video event respectively, and let $M_\theta$ represent the TSG model, the goal of TSG task can be formulated as $M_\theta(v, q; V, Q, L) \rightarrow \{(t_s^k, t_e^k)\}_{k=1}^K, t_s^k < t_e^k$. $(t_s, t_e)_k$ is the $k$-th retrieved moment in the ranked list of candidates. $K$ is the number of predicted candidates. $L$ represents the manual temporal annotations of training sets. Particularly, let $L_{iid}$ and $L_{ood}$ represent the ground truth of two test datasets sharing identical distribution and outlier distribution with the training set $L$, $M_\theta$ is expected to achieve promising performance on both test datasets synchronously, which requires better generalization ability.

**Causal Graph in Our CAEM Method.** The causal graph is a directed acyclic graph that reveals the causal relationships among variables. The nodes denote the variables and the edges represent the causal relationships among variables. As the proposal-based pipeline of TSG methods firstly generates video event candidates and then measures matching scores between each event and text queries, we describe causalities among five variables: text query $Q$, video event $E$, corresponding temporal location in the video $L$, the joint event-query representations $M$ and predicted matching score $S$. Concretely, we illustrate the causal graph in Fig. 1(c) to show how the variables $\{Q, E, L, M, S\}$ interact with each other through the

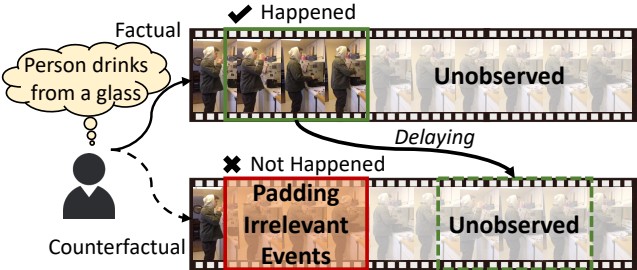

**Figure 3: An illustration of our Temporal Counterfactual Augmentation: at the annotated moment, we delay the matched video events to those video parts that have not been observed, thus generating a counterfactual that the event-query matching does not happen currently but later.**

causal links, where the direct link denotes the causality between two nodes: cause → effect. The temporal location in the video $L$ will influence indirectly the predicted matching scores $S$ between text queries $Q$ and video events $E$ via the joint representations $M$.

## 3.2 Temporal Counterfactual Augmentation

**Discussions on Factual Event-Query Matching Annotations.** Given a video that contains multiple events, it can be regarded as a time series of visual signals, and its semantics are aligned with the textual modalities along the temporal dimension. Such a priori condition leads to directed contextual information, thus we humans tend to discriminate if an event happens by considering current situations and previous memories. For example, we illustrate a typical case in Fig. 3. For any annotator, who is asked to determine the starting and ending timestamp with the given text query, it firstly watches the video until the described event begins at this time, and then keeps recording in mind until it ends. In this way, an annotated moment is generated, which represents an observed fact that the required event happened at this moment and the query is expected to build a strong correlation with this video content.

**Counterfactual Data Generation.** In the context of the discussions above, we find that humans don't have to observe the rest of this video to discriminate the matching relationships between video events and textual queries. Therefore, we argue that the event-query matching should be context-irrelevant since the semantics video event is complete enough to build flawless matching pairs with text queries. As illustrated in Fig. 4, we raise counterfactual thinking in determining if the event happens at a particular moment: *What if the event is not observed now but later?* Under the counterfactual consistency rule [13, 14], if the event does not happen at the current moment but still happens in the unobserved rest part of the video, the matching correlations should be established consistently.

To this end, we incorporate counterfactual thinking in data augmentation and propose a novel strategy for counterfactual data generation. Specifically, we denote any annotated video-text pair selected from the training set as $(v, q, l)$, where $l = (t_s, t_e)$ is the normalized moment, and denote the lengths of video as $l_v$. For short events which satisfy $l/l_v < 0.5$, we move the event frames entirely and insert them at $t_s + \rho$ with a random ratio $\phi$, delaying the annotated event to $(t_s + \rho_1, t_e + \rho_1)$. Here $\rho_1 \in [0, 1 - t_e]$ is a random relative temporal distance. Moreover, for those events that

satisfy $l/l_v \geq 0.5$, we pad a group of video frames with complete event semantics at the beginning of the video sample, extending the video length into $(1 + \rho_2) * l_v$, where $\rho_2 \in [0, 3 * (t_e - t_s)]$ is a random relative temporal distance similarly. Here we also conduct the extending operation with a random ratio $\phi$. Particularly, the padded video frames are selected from the training set, which is a complete event from another video. In this way, we can get a group of counterfactually augmented training pairs, which can be used to train our models with original factual data jointly.

## 3.3 Event-Query Matching Model

**Feature Extraction.** Following the previous methods [5, 10, 34], we extract the video and text features in an offline manner. Firstly, given an untrimmed video $V$, we adopt the pre-trained visual backbones [48, 49] to extract the visual features. Moreover, we employ the GloVe [50] to extract the word-level embeddings of each sentence query $Q$. Specifically, we apply mean pooling on the frame-level representations within each clip to obtain video feature sequence, represented as $\mathbf{F}_V = \{\mathbf{f}_V^i | i = 1, 2, ..., l_V\}$, where $\mathbf{f}_V^i \in \mathbb{R}^{d_V}$ denote the $i$-th clip feature and $l_V$ is the video feature length. For the query sentence $Q$, the feature sequence consists of multiple word-level embeddings, denoted as $\mathbf{F}_Q = \{\mathbf{f}_Q^j | i = 1, 2, ..., l_Q\}$, where $\mathbf{f}_Q^j \in \mathbb{R}^{d_Q}$ and $l_Q$ are the $j$-th word feature and the number of words respectively.

**Event-Query Joint Representation.** To learn an effective joint representation for event-query pairs, we incorporate vision-language transformers [51] into temporal adjacent networks [3]. Concretely, the vision-language transformers are deployed to conduct cross-modal interactions thus learning fine-grained multimodal representations that contain both visual and textual information. Furthermore, we leverage the temporal adjacent map [3] to generate abundant candidates, where the representation of each candidate is composed of multiple multimodal tokens sampled from the output of vision-language transformers. Let the VLTrans $(\cdot)$ denote vision-language transformers, we formulate this process as follows:

$$\mathbf{E}_V, \mathbf{E}_Q = \text{VLTrans}(\mathbf{F}_V, [\mathbf{h}_{\text{cls}}; \text{Mask}(\mathbf{F}_Q)]), \quad (1)$$

where Mask $(\cdot)$ is a random mask operation: we adopt masked language modeling [51] here to learn more semantics from textual queries. $\mathbf{h}_{\text{cls}}$ is the *<CLS>* token and [; ] is the concatenation operation. $\mathbf{E}_V = \{\mathbf{e}_V^i | i = 1, 2, ..., l_V\}$ and $\mathbf{E}_Q = \{\mathbf{e}_{\text{cls}}, \mathbf{e}_Q^i | i = 1, 2, ..., l_Q\}$ are vision-aware and text-aware multimodal token sequence respectively, $\mathbf{e}_{\text{cls}}, \mathbf{e}_V^i, \mathbf{e}_Q^i \in \mathbb{R}^{d_h}$.

Following that, we generate a temporal adjacent map [3] illustrated in Fig. 2, and fill each grid with composed features sampled from the vision-aware multimodal tokens. In this temporal adjacent map, the grid in $(t_i, t_j)$ represents a candidate event starting at $t_i$ and ending at $t_j$ timestamps in the untrimmed videos, where $0 <= t_i <= t_j <= l_V$. Take the grid $(t_i, t_j)$ as an example, we denote the process of filling multimodal features as follows:

$$\mathbf{e}_M^{(t_i, t_j)} = \mathbf{W}_1 \mathbf{e}_V^{t_i} + \mathbf{W}_2 \mathbf{e}_V^{t_j} + \mathbf{W}_1 \mathbf{e}_V^{\lfloor (t_i + t_j)/2 \rceil} + \mathbf{b}, \quad (2)$$

where $\mathbf{W}_{1,2,3}$ are learnable parameter matrics and $\lfloor \cdot \rceil$ represents the rounding operation. $\mathbf{b}$ is the bias matrix. $\mathbf{e}_M^{(t_i, t_j)}$ is the event-query joint representation for a candidate event at $(t_i, t_j)$.

**Semantic Consistency Learning.** As the matched event-query pairs share the same semantic in both vision and text modalities, we firstly leverage cross-modal contrastive learning to guide our model to learn semantic consistency. Specifically, given the feature of the target event $\mathbf{e}_M^+$, and the global text-aware token $\mathbf{e}_{cls}$, we project these features into a joint embedding space and let the model distinguish between anchor sample and negative samples. Let $N_v$ represent a set of features containing $\mathbf{e}_M^+$ and the negative samples of mis-matched event-query pairs from the same and other videos, the contrastive learning loss can be formulated as follows:

$$L_{cl} = \mathbb{E}_{\sim p(\Sigma)} \left[ -\log \left( \frac{\exp \left( S \left( f_v \left( \mathbf{e}_M^+ \right) f_q \left( \mathbf{e}_{cls} \right) \right) / \tau \right)}{\sum_{\kappa \in N_v} \exp \left( S \left( f_v \left( \mathbf{e}_M^\kappa \right) f_q \left( \mathbf{e}_{cls} \right) \right) / \tau \right)} \right) \right], \quad (3)$$

where $S(\cdot)$ is the cosine similarity calculation. $p(\Sigma)$ represents the expected distribution depicting the correlations of $\{V, Q, L\}$. $\tau$ is a temperature factor. $f_v$ and $f_q$ are project layers with normalizing operations for visual and textual modalities respectively.

Additionally, we also employ the widely used Masked Language Modeling (MLM) to exploit the semantics hidden in textual queries. Given the text-aware multimodal token sequence $\mathbf{E}_Q$, we predict the masked words in $\text{Mask}(\mathbf{F}_Q)$. Take the $i$-th word masked as an example, the training loss for MLM is denoted as follows:

$$L_{mlm} = \mathbb{E}_{\sim p(\Sigma)} \left[ -\log p(w_i | w_1, w_2, ..., w_{i-1}, w_{i+1}, ..., w_{l_Q}) \right], \quad (4)$$

where $w_i$ represents $i$-th word at the original textual query sentence. Overall, the loss function of the Semantic Consistency Learning can be summarized as $L_{SCL} = L_{cl} + L_{mlm}$.

**Temporal Supervision Perception.** We devise two parts of loss functions to perceive direct supervision signals from the manual annotations, i.e., matching confidence and relative location modification. The first part is used to guide our model to discriminate whether an event-query pair, which has been represented as composed multimodal features, is well-matched. Moreover, as the temporal adjacent map is pre-defined manually, we also incorporate the second part, i.e., relative location modification to enable our model to predict a subtle relative location for each grid. Specifically, the matching confidence can be calculated as follows:

$$L_{ms} = \mathbb{E}_{\sim p(\Sigma)} \left[ \mathbb{E}_{\sim p(S|\Sigma)} \left[ (-\hat{s}_i \log s_i - (1 - \hat{s}_i) \log (1 - s_i)) \right] \right], \quad (5)$$

where $s_i = f_s(\mathbf{e}_M^i)$ is a predicted matching score of the $i$-th candidate event in the temporal adjacent map, and $f_s$ is a classifier. Meanwhile, $\hat{s}_i$ is a ground truth matching score of the $i$-th candidate event, which is generated by calculating the temporal IoU score between the pre-defined temporal adjacent map and manual annotations, such as:

$$\hat{s}_{(t_s^i, t_e^i)} = \frac{\min(\hat{t}_e, t_e^i) - \max(\hat{t}_s, t_s^i)}{\max(\hat{t}_e, t_e^i) - \min(\hat{t}_s, t_s^i)}, \quad (6)$$

where $(t_s^i, t_e^i)$ represents normalized timestamps in the $i$-th grid at the temporal adjacent map and ground truth, while $(\hat{t}_s, \hat{t}_e)$ is the normalized ground truth. Similarly, the second part loss used for relative location modification can be formulated as:

$$L_{rlm} = \mathbb{E}_{\sim p(\Sigma)} \left[ \mathbb{E}_{\sim p(S|\Sigma)} \left[ \left| r_s^i + t_s^i - \hat{t}_s \right| + \left| r_e^i + t_e^i - \hat{t}_e \right| \right] \right], \quad (7)$$

where $(r_s^i, r_e^i) = f_r(\mathbf{e}_M^i)$, which is the margin distance between the pre-defined location $(t_s^i, t_e^i)$ and ground truth location $(\hat{t}_s, \hat{t}_e)$.

Here $f_r$ is a regression head for predicting margin distance. Overall, the training objective for perceiving temporal supervision can be summarized as: $L_{TSP} = L_{ms} + L_{rlm}$.

## 3.4 Counterfact-Adaptive Framework

Training the Event-Query Matching Model with our counterfactually augmented video-text pairs could enhance the counterfactual thinking ability, but the intervened video frame order may also show influence in vision-language transformers as the changed token sequences. To this end, we furtherly propose the Counterfact-Adaptive Framework to overcome this defect.

**Pseudo-Siamese Transformers.** As the goal of our Counterfact-Adaptive Framework is to mitigate the influence caused by the changed token sequences in transformers, we firstly extend the Vision-Language Transformers into a pseudo-siamese architecture [52]. As illustrated in Fig. 2, we employ two groups of vision-language transformers with the same architecture but unshared parameters. They are designed to handle different inputs: one is asked to process the original training data while the other one aims to process the counterfactually augmented data. Specifically, let $\mathbf{F}_V^*$ represent the counterfactual augmented video features, we denote the pseudo-siamese vision-language transformers as follows:

$$\begin{cases} \mathbf{E}_V, \mathbf{E}_Q = \text{VLTrans}(\mathbf{F}_V, [\mathbf{h}_{cls}; \text{Mask}(\mathbf{F}_Q)]), \\ \mathbf{E}_V^*, \mathbf{E}_Q^* = \text{VLTrans}^*(\mathbf{F}_V^*, [\mathbf{h}_{cls}^*; \text{Mask}(\mathbf{F}_Q)]), \end{cases} \quad (8)$$

where "*" denotes the pipeline for counterfactual augmented data. We employ two group of unshared learnable matrices, i.e., $\mathbf{W}_{1,2,3}$ and $\mathbf{W}_{1,2,3}^*$ to fill multimodal features for each candidate in two pipeline respectively. Similarly, we also calculate the Semantic Consistency Learning loss for two pipelines separately, i.e., $L_{SCL}$ and $L_{SCL}^*$. Finally, we employ a *shared* classifier to predict matching score for each candidate for both two vision-language transformers and combine them to learn final matching scores. Similarly, take the $i$-th candidate event as an example, the process of predicting the final matching score can be denoted as: $s_i' = f_s(\mathbf{e}_M^i) + \lambda f_s(\mathbf{e}_M^{i*})$, where $\mathbf{e}_M^i$ and $\mathbf{e}_M^{i*}$ are the multimodal features of $i$-th candidate event in the two predicted temporal adjacent map. $s_i'$ and $f_s(\cdot)$ are predicted final matching score and shared $f_s(\cdot)$ classifier.

**Counterfactual Consistency Learning.** Following the counterfactual consistency rule [13, 14], if the event does happen in the rest of the videos, the event-query pairs should also be well-matched. In other words, the completeness of semantics in the same video events should be maintained. To this end, we leverage contrastive learning to align the multimodal representations of the same event from both two pipelines. Specifically, let $\hat{\mathbf{e}}_M$ and $\hat{\mathbf{e}}_M^*$ represent the multimodal representations for ground truth events in factual and counterfactual pipelines, the loss function of counterfactual consistency learning can be represented as:

$$L_{CCL} = \mathbb{E}_{\sim p(\Sigma)} \left[ -\log \left( \frac{\exp \left( S \left( f_v \left( \hat{\mathbf{e}}_M^* \right) f_v \left( \hat{\mathbf{e}}_M^+ \right) \right) / \tau \right)}{\sum_{\hat{\mathbf{e}}_M^n \in N_e} \exp \left( S \left( f_v \left( \hat{\mathbf{e}}_M^* \right) f_v \left( \hat{\mathbf{e}}_M^n \right) \right) / \tau \right)} \right) \right]. \quad (9)$$

Here $\hat{\mathbf{e}}_M^+$ and $\hat{\mathbf{e}}_M^n$ represent multimodal representations of the same and different events respectively. $N_e$ represent a set of features containing $\hat{\mathbf{e}}_M^+$ and negative samples. $\tau$ is the temperature factor.

## 4 Experiments

### 4.1 Experimental Settings

**Datasets.** We evaluate our proposed CAEM models on the *Charades-CD* and *ActivityNet-CD* [5] datasets, which are repartitioned to evaluate the performance and generalizing ability of TSG models. In the two datasets, the target moments of samples in the training, val, and test-iid sets are independent and identically distributed (denoted as *IID*), while the test-ood set contains out-of-distribution (denoted as *OOD*) samples. Moreover, we also generalize our proposed CAEM method on the *Charades-CG* and *ActivityNet-CG* [53] datasets, which contain normal test queries where all words are seen in the training set (denoted as *Trivial*) and generalized test queries with unseen words in the training set (denoted as *Novel*).

**Implementation Details and Metrics.** Following the previous methods [3, 5, 11, 34], we adopt the off-the-shelf video features that are extracted by pre-trained 3D CNN backbones [48, 49]. The visual features and textual embeddings are projected into 256 dimensions before sending to vision-language transformers, and the hidden dimension of transformers is also set to 256. As for the hyperparameters, we set the random ratio $\phi$ in the Temporal Counterfactual Augmentation to 0.5. The balance factor $\lambda$ for calculating the final matching score in the Counterfact-Adaptive Framework is set to 0.6. The temporal factor in $L_{SCL}$ and $L_{CCL}$ is set to 0.1. the training stage, we employ the Adam optimizer [54] is employed to update the parameters with the learning rate set to $4 \times 10^{-4}$ on a single Nvidia A6000 with 64 batch size. During the evaluation and testing stages, we adopt the widely used de-biased recall metrics proposed by [5], which evaluate the model generalization on OOD samples by discounting the normal recall metrics for suppressing the performance of speculation methods that over-rely on moments annotation biases. Moreover, as a number of D-TSG methods [9, 11, 12, 34] employed conventional recall metrics to evaluate the model performance on the D-TSG task, we also evaluate our model under the same metrics for fair comparisons.

### 4.2 Overall Comparision

We compare our proposed CAEM method with recent state-of-the-art methods [5, 6, 10, 11, 34, 55]. For fairness, we adopt two types of metrics, *i.e.*, De-Biased Metrics [5] and Normal Metrics [1, 7], covering all previous methods evaluated with different metrics. Specifically, we summarize the counterpart methods as following two groups: (1) *De-Biased Metrics:* TCN-DCM [6], MDD [5], Multi-NA [10], and DFM [11]. (2) *Normal Metrics:* D-TSG [12], SVTG [9], MomentDiff [55], and BSSARD [34]. To further show the effectiveness of our method, we also make fair comparisons with the latest methods that show generalization on text modality on the Charades-CG and ActivityNet-CG [53], including VISA [53], VDI [56], SSL2CG [57], and MESM [58].

**Comparisons with Recent DTSG Methods.** We report the experimental results on Charades-CD and ActivityNet-CD [5] in Table 1 and Table 2. By comparing our proposed CAEM method and the recent state-of-the-art methods, we list the following observations: (1) Our method consistently outperforms these counterpart methods on both benchmark datasets. Particularly, on the OOD test of the Charades-CD dataset, our method achieves 4% and 7% absolute improvements on the de-biased discounted metric dR@1, IoU=0.5,

**Table 1: Comparisons with recent state-of-the-art DTSG methods on the Charades-CD datasets.**

| Method | Venue | dR@1, IoU=0.5 | | dR@1, IoU=0.7 | |
|---|---|---|---|---|---|
| | | IID | OOD | IID | OOD |
| VSLNet [4] | ACL'20 | 47.60 | 32.72 | 29.88 | 19.61 |
| 2D-TAN [3] | AAAI'20 | 46.48 | 30.77 | 28.76 | 13.73 |
| TCN-DCM [6] | SIGIR'21 | 52.50 | 40.51 | 35.28 | 21.02 |
| MDD [5] | TOMM'22 | 52.78 | 40.39 | 34.71 | 22.70 |
| Multi-NA [10] | AAAI'23 | 53.82 | 39.86 | 34.47 | 21.38 |
| DFM [11] | ACM MM'23 | 56.38 | 44.01 | 34.87 | 22.28 |
| CAEM | Ours | **63.38** | **48.72** | **43.76** | **26.36** |
| Method | Venue | R@1, IoU=0.5 | | R@1, IoU=0.7 | |
| | | IID | OOD | IID | OOD |
| D-TSG [12] | ACM MM'22 | 53.34 | 41.42 | 31.35 | 22.04 |
| SVTG [9] | ECCV'22 | 57.59 | 46.67 | 37.79 | 27.08 |
| MomentDiff [55] | NeurIPS'23 | - | 47.17 | - | 22.98 |
| BSSARD [34] | AAAI'24 | 55.65 | 47.20 | 36.33 | 27.17 |
| CAEM | Ours | **68.53** | **54.42** | **46.17** | **28.29** |

**Table 2: Comparisons with recent state-of-the-art DTSG methods on the ActivityNet-CD datasets.**

| Method | Venue | dR@1, IoU=0.5 | | dR@1, IoU=0.7 | |
|---|---|---|---|---|---|
| | | IID | OOD | IID | OOD |
| VSLNet [4] | ACL'20 | 39.86 | 19.57 | 26.45 | 11.14 |
| 2D-TAN [3] | AAAI'20 | 40.87 | 18.86 | 27.36 | 9.77 |
| TCN-DCM [6] | SIGIR'21 | 42.15 | 20.86 | 29.69 | 11.07 |
| MDD [5] | TOMM'22 | 43.63 | 20.80 | 31.44 | 11.66 |
| Multi-NA [10] | AAAI'23 | 41.67 | 20.78 | 28.82 | 11.03 |
| DFM [11] | ACM MM'23 | 45.92 | 24.32 | 32.18 | 12.72 |
| CAEM | Ours | **46.07** | **26.42** | **33.92** | **14.80** |
| Method | Venue | R@1, IoU=0.5 | | R@1, IoU=0.7 | |
| | | IID | OOD | IID | OOD |
| SVTG [9] | ECCV'22 | 48.07 | 24.57 | 32.15 | 13.21 |
| MomentDiff [55] | NeurIPS'23 | - | 26.96 | - | 13.69 |
| BSSARD [34] | AAAI'24 | 49.67 | 27.02 | 33.72 | 14.93 |
| CAEM | Ours | **51.55** | **28.98** | **36.36** | **15.54** |

and conventional metric R@1, IoU=0.5 compared with existing state-of-the-arts. It demonstrates that our method shows better generalization towards OOD samples. This is because our proposed CAEM method leverages the TCA and CAF designs, which enhance counterfactual thinking in learning joint representations of matched query-event pairs, thus showing better generalization compared with others. (2) By comparing the improvements achieved by our method on the Charades-CD and ActivityNet-CD datasets, it can be observed that our proposed CAEM method shows more superiority over the former benchmark dataset. One probable reason is the described events in the ActivityNet-CD dataset contain more complicated semantics. These complex queries raise more challenges in learning joint representations for matched event-query pairs.

**Model Generalization on Novel Queries.** We generalize our proposed CAEM method on two additional TSG methods, *i.e.*, Charades-CG and ActivityNet-CG, which contain novel words in test queries that are unseen in the training set. According to the experimental results in Table 3, we can observe that our proposed CAEM methods also achieve superior results and outperform existing methods with remarkable margins. Specifically, compared with the latest state-of-the-art method SSL2CG [57], which aims to improve the

generalization of models towards novel word or word-level compositions via self-supervised contrastive learning, our method achieves approximate 2% absolute improvements at least on all the metrics. Such results demonstrate that our proposed CAEM methods also show superior generalization toward text queries with unseen words. This is because our method focuses on the causal structure of learning joint representations with textual queries, events, and temporal locations, instead of processing a single modality or predicting grounding results directly.

**Table 3: Experimental results with novel words in text queries on the Charades-CG and ActivityNet-CG datasets.**

| Method | Venue | R@1, IoU=0.5 | | R@1, IoU=0.7 | |
|---|---|---|---|---|---|
| | | Trivial | Novel | Trivial | Novel |
| | Charades-CG | | | | |
| VSLNet [4] | ACL'20 | 45.91 | 25.60 | 19.80 | 10.07 |
| 2D-TAN [3] | CVPR'20 | 48.58 | 29.36 | 26.49 | 13.21 |
| VISA [53] | CVPR'22 | 53.20 | 42.35 | 26.52 | 20.88 |
| VDI [56] | CVPR'23 | - | 46.47 | - | 28.63 |
| SSL2CG [57] | CVPR'23 | 58.14 | 50.36 | 37.98 | 28.78 |
| MESM [58] | AAAI'24 | - | 50.50 | - | 33.67 |
| CAEM | Ours | **62.27** | **54.10** | **39.44** | **35.97** |
| | ActivityNet-CG | | | | |
| VSLNet [4] | ACL'20 | 39.27 | 21.68 | 23.12 | 9.94 |
| 2D-TAN [3] | CVPR'20 | 44.50 | 23.86 | 26.03 | 10.37 |
| VISA [53] | CVPR'22 | 47.13 | 30.14 | 29.64 | 15.90 |
| VDI [56] | CVPR'23 | - | 32.35 | - | 16.02 |
| SSL2CG [57] | CVPR'23 | 49.63 | 30.15 | 31.73 | 14.97 |
| CAEM | Ours | **51.99** | **32.52** | **34.36** | **16.45** |

## 4.3 Further Analysis

**Ablation Studies.** To explore the impact of different model components in our proposed CAEM method, we conduct ablation studies on the three key modules. Specifically, the experimental results are illustrated in Table 4, and we can observe that: (1) The SCL performs a crucial role for IID test samples. By comparing the second and third lines in the upper subtable, we find that the ablated model *SCL + TCA* achieves remarkable improvements over the ablated model *TCA*. Moreover, the comparisons between *TCA + CAF* and the complete model *SCL + TCA + CAF* also illustrate the effectiveness of SCL. This is because the SCL is used to guide our model to learn hidden semantics in text queries and video events, and further align them within a common space. (2) The TCA significantly mitigates the bias in the training set and enhances the generalization ability of our models toward out-of-distribution test samples. We can see that the ablated models with TCA consistently achieve remarkable improvements on the OOD test results, as the TCA is used to teach models counterfactual thinking. (3) Moreover, we can also note that the CAF effectively boosts the comprehensive performance on both IID and OOD test samples. Facilitated with the CAF module, the models achieve the best performance on most metrics and outperform other ablated models with remarkable margins.

**Analysis on Temporal Counterfactual Augmentation.** To furtherly demonstrate the effectiveness of our proposed TCA module, we conduct more analysis on the random ratio $\phi$ which controls

**Table 4: Experimental results of ablation studies on the Charades-CD datasets.**

| SCL | TCA | CAF | dR@1, IoU=0.3 | | dR@1, IoU=0.5 | | dR@1, IoU=0.7 | |
|---|---|---|---|---|---|---|---|---|
| | | | IID | OOD | IID | OOD | IID | OOD |
| ✓ | ✗ | ✗ | 67.60 | 56.20 | 58.95 | 44.37 | 41.85 | 23.99 |
| ✗ | ✓ | ✗ | 67.04 | 59.32 | 58.86 | 48.00 | 41.55 | **27.47** |
| ✓ | ✓ | ✗ | 68.56 | 58.69 | 61.34 | 46.34 | 42.18 | 26.12 |
| ✗ | ✓ | ✓ | 69.56 | 59.80 | 62.28 | 48.01 | 41.81 | 26.04 |
| ✓ | ✓ | ✓ | **70.22** | **60.57** | **63.38** | **48.72** | **43.76** | 26.36 |

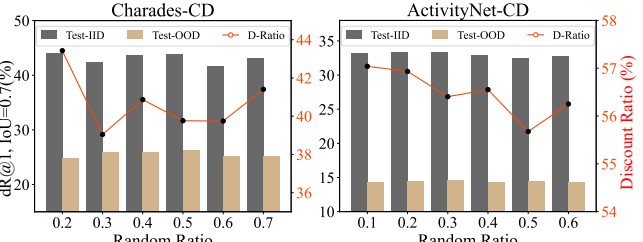

**Figure 4: Analysis for the influence on model performance of Discount Ratio Δ (denoted as D-Ratio) and of Random Ratio $\phi$ in TCA on Charades-CD and ActivityNet-CD datasets.**

if a sample will be transformed to a counterfactual case. Specifically, we adjust $\phi$ from low to high and observe how the model performance on OOD test samples fluctuates. Moreover, here we also adopt a discount ratio between the test results of IID and OOD sets to quantify the general performance, which could be calculated as: $\Delta = (R_{\text{IID}} - R_{\text{OOD}})/R_{\text{IID}}$. Here $R_{\text{IID}}$ and $R_{\text{OOD}}$ represent model performance on IID and OOD test samples respectively. Note that lower $\Delta$ means better general performance of a DTSG model.

By observing the experimental results illustrated in Fig. 4, we can see that our model achieves the best performance on OOD test samples when the random ratio $\phi$ is near 0.5. Moreover, with a low or over-high random ratio $\phi$, the performance drops dramatically. One probable reason is the delicate balance between factuals and counterfactuals: A low random ratio undermines the effectiveness of TCA while an over-high random ratio increases the difficulty of exploiting the counterfactual consistency rule in CAF, thus both lead to poor generalization toward OOD samples.

**Analysis on Counterfact-Adaptive Framework.** We also conduct further analysis of the CAF module to show its superiority. Specifically, we follow the experiments adopted in the analysis of TCA to explore how the performance of models facilitated by CAF fluctuates with the random ratio $\phi$. Moreover, we also observe the influence of the balance factor $\lambda$ by evaluating the improvements of *dR@1, IoU=0.5* achieved by our complete CAEM over *CAEM w/o CAF*. The experimental results above are illustrated in Fig. 4 and Fig. 5 respectively. In the context of the illustrations, we can observe that: (1) The CAF significantly boosts the effectiveness of our proposed TCA module. By comparing the ablated model *CAEM w/o CAF* and the complete model *CAEM*, we find the CAF consistently brings better improvements under different random ratio conditions. It demonstrates the superiority of introducing the counterfactual consistency rule into our models for learning semantic-invariable matching between video events and textual queries. (2) Higher balance factor $\lambda$ can lead to a degeneration of general performance.

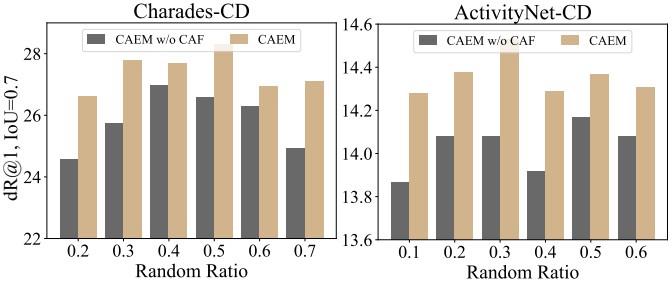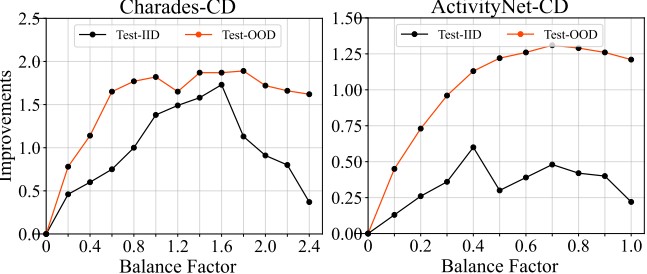

**Figure 5: Further analysis w.r.t. CAF module on the Charades-CD and ActivityNet-CD datasets, including model performance under different Random Ratios $\phi$ and the performance improvements brought by CAF under different balance factors $\lambda$.**

With $\lambda$ increasing, the IID performance is decreased while the OOD performance is improving gradually. We speculate that this is due to models increasingly emphasizing counterfactual consistency at the expense of overall model flexibility. When $\lambda$ is set too high, the model becomes overly focused on imaging counterfactual instances, at the cost of learning meaningful semantic representations and capturing diverse patterns in the data.

**Table 5: Analysis with respect to the training objectives in Event-Query Matching Model on the Charades-CD datasets.**

| $L_{\mathrm{rlm}}$ | $L_{\mathrm{mlm}}$ | $L_{\mathrm{cl}}$ | dR@1, IoU=0.3 | | dR@1, IoU=0.5 | | dR@1, IoU=0.7 | |
|---|---|---|---|---|---|---|---|---|
| | | | IID | OOD | IID | OOD | IID | OOD |
| ✗ | ✗ | ✗ | 67.27 | 57.78 | 57.64 | 45.37 | 39.18 | 25.69 |
| ✓ | ✗ | ✗ | 69.56 | 59.80 | 62.28 | 48.01 | 41.81 | 26.04 |
| ✓ | ✓ | ✗ | 70.12 | 59.16 | 62.91 | 46.41 | 41.49 | 26.25 |
| ✓ | ✗ | ✓ | 68.19 | 60.24 | 59.94 | 48.22 | 42.32 | 26.22 |
| ✓ | ✓ | ✓ | **70.22** | **60.57** | **63.38** | **48.72** | **43.76** | **26.36** |

**Analysis on Event-Query Matching Model.** We also take a closer look at the Event-Query Matching (EQM) model by conducting further ablation studies on the training objectives, including relative location modification loss $L_{\mathrm{rlm}}$, masked language modeling loss $L_{\mathrm{mlm}}$, and cross-modal contrastive learning loss $L_{\mathrm{cl}}$. According to the experimental results in Table 4, we list the following observations: (1) $L_{\mathrm{rlm}}$ performs an essential role in our EQM model for both IID and OOD performance. We can see that the model performance on both IID and OOD drops dramatically without $L_{\mathrm{rlm}}$. This is because the representations of event candidates in our EQM are learned with the compositions of vision-aware tokens, which have fixed mapping temporal locations determined manually. $L_{\mathrm{rlm}}$ aims to teach our model to learn relative temporal locations between these fixed temporal locations and ground truth. (2) Both $L_{\mathrm{cl}}$ and $L_{\mathrm{mlm}}$ show effectiveness in model performance. This is because the two training objectives are designed to exploit the hidden semantics and learn cross-modal alignment for better joint representations.

**Qualitative Analysis.** We illustrate two typical test cases from the Charades-CD and ActivityNet-CD datasets in Fig. 6. Moreover, to show the generalization of DTSG models, we also visualize the distributions of the training set following [10, 11]. By observing the visualization results, we can see that our proposed CAEM method precisely localizes the target event that is the most relevant to the given text query. Specifically, compared with the state-of-the-art DTSG method BSSARD [34], our CAEM method achieves more

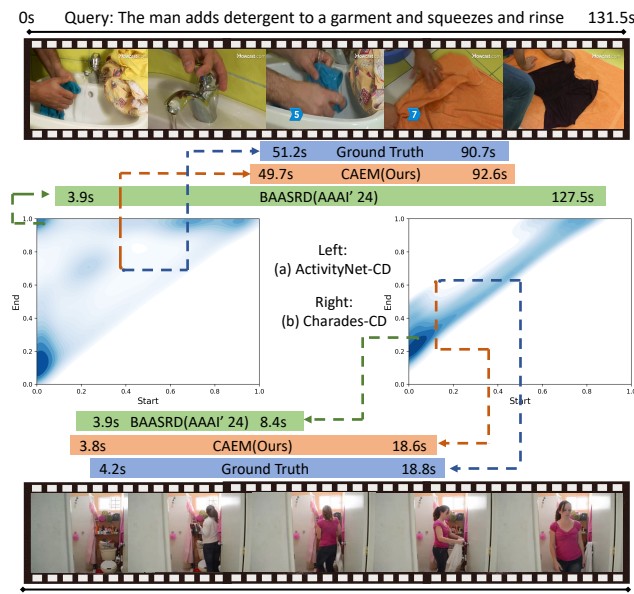

**Figure 6: Visualizations of grounding results of OOD test samples from (a) ActivityNet-CD and (b) Charades-CD datasets and corresponding distributions of training sets.**

accurate grounding results. Moreover, by observing the distributions illustrated in Fig. 6(a) and (b), we can also find that our CAEM method shows better generalization on OOD test samples. It proves the effectiveness of our solution again, which introduces counterfactual data augmentation and consistency rule into event-query matching to achieve better generalization.

## 5 Conclusion

In this paper, we proposed a novel method, termed Counterfactually Augmented Event Matching (CAEM), for the De-biased Temporal Sentence Grounding (DTSG) task. It aims to improve OOD generalization in localizing video events with given text queries by introducing counterfactual data augmentation and consistency rule into event-query matching. Thorough experiments on three benchmark datasets demonstrated our proposed method established new state-of-the-art performance on the DTSG task. For future work, we will study the bias in multimodal learning, and explore corresponding solutions for causal reasoning.

## 6 Acknowledgments

This work was supported by the National Natural Science Foundation of China under Grants (No. 62222203 and 62072080), the New Cornerstone Science Foundation through the XPLORER PRIZE and the Sichuan Science and Technology Program (No. 2023-XT00-00001-GX).

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
