# OpenReview forum: "Counterfactually Augmented Event Matching for De-biased Temporal Sentence Grounding"
_acmmm.org/ACMMM/2024/Conference — MM2024 Poster_

### Official Review · Reviewer_6yho · 2024-05-06

**Rating:** 5
**Confidence:** 3

**Summary:**

This paper addresses the out-of-distribution problem between train and test set in temporal sentence grounding. In this paper, they design a novel framework, termed Counterfactually-Augmented Event Matching. They devise 3 novel components to help the model to mitigate the distribution bias in datasets. The results on benchmarks shows the effecience.

**Strengths:**

1. It introduces a novel debiased network for TSG. It devises a new counterfactual augmentation method by delaying the target moment and proposes counterfact-adaptive framework to learn event-query joint representations. The modules they proposed appear to be reasonable and complementary. Experiments on benchmarks seem to show the effectiveness.
2. Extensive experiments is sufficient and convincing.
Overall, this is a complete and novelty works.

**Limitations:**

1.  Temporal Counterfactual Augmentation generates the counterfactual samples by delaying the target video segment and insert a random event into the previous location. However, this approach may result in moving backwards of distribution of target moment , which seems to not satisfy the independence of variable in counterfactual reasoning. It needs some extension experiments or at least a disscussion about it.
2. There are many losses in training, sensitive analysis of their weight balance is needed.

**Suitability:**

3

---

### Official Review · Reviewer_ub5m · 2024-05-24

**Rating:** 3
**Confidence:** 4

**Summary:**

The paper introduces a novel framework called Counterfactually Augmented Event Matching (CAEM) to address training bias through counterfactual data augmentation.The framework includes a Temporal Counterfactual Augmentation module, an Event-Query Matching mode and a Counterfact-Adaptive Framework (CAF) that applies counterfactual consistency rules to the matching process, further mitigating training set biases. The experimental results demonstrate CAEM's potential in improving event-query representation learning and reducing bias.

**Strengths:**

1. This paper addresses the issue of dataset bias in TSG, presenting an intuitive and effective solution.
2. The proposed method's efficacy is validated through experiments on two types of debiased datasets.

**Limitations:**

1. This paper only performs comparisons on debiased datasets. While ActivityNet-CD and Charades-CD contain IID parts, it is recommended that the authors also conduct experiments on general datasets to compare performance in IID scenarios.
2. The overall framework of this paper is based on a proposal method. Given the prevalence of proposal-free methods, it would be beneficial to explore the adaptation of this approach to a proposal-free framework.

**Suitability:**

3

---

### Official Review · Reviewer_gVJg · 2024-05-25

**Rating:** 3
**Confidence:** 2

**Summary:**

This paper tackles the task of debiased video grounding. It proposes a counter-factual augmentation module, an event-query matching model, and a counter-factual adaptive framework. Experiments on Charades-STA and ActivityNet-Captions demonstrate the effectiveness of the proposed methods.

**Strengths:**

1. The paper is well-organized, particularly the presentation of experiments, which is structured effectively and ensures a smooth reading experience.
2. The experiments conducted on two datasets demonstrate the effectiveness of the proposed methods.

**Limitations:**

**Major concerns:**

1. The motivation of proposed method still presents remains unclear.

    i) Why does the independence of the event, queries, and location contribute to debiasing grounding?

    ii) In my understanding, bias in grounding is mainly caused by the bias in annotated location. Given this, a debiased grounding model should: 1) be based solely on the matching of video content and text semantics, and 2) eliminate the interference of location when determining the matching score. According to Figure 1c, location information is explicitly injected into the multimodal representation. How does this result in debiased grounding?

2. Why does the author claim that the event, queries, and location are independent in their method? Conversely, why are these three entities dependent in conventional methods? In conventional grounding, the multimodal representation is also determined by these three entities independently (e.g., the previous method 2D-TAN).
3. How are Figures 1c and 2 related? Figure 1c is the causal graph of the proposed method, while Figure 2 shows the details of the proposed method. How is the idea in Figure 1c instantiated in Figure 2?  Additionally, Figure 2 indicates that the main framework of the proposed method is still based on a 2D Map, a framework from the previous 2D-TAN (which is designed for common grounding). Given this, why does the author claim their method is fundamentally different as claimed in Figure 1?

**Other concerns:**

- According to Line 658, dR@1, IoU=0.5 is used to denote the debiased discounted metric, while R@1, IoU=0.5 represents the conventional metric. Why are the results for conventional  2D-TAN [3] reported under dR@1, IoU=0.5 instead of R@1, IoU=0.5?

**Note:** While I have concerns about the ambiguity of the motivation, I find the overall structureof the paper to be good and clear. I would raise my final rating if the author's response addresses my major concerns.

**Suitability:**

3

---

### Official Review · Reviewer_t8zf · 2024-06-03

**Rating:** 4
**Confidence:** 3

**Summary:**

The paper proposes a novel framework for addressing bias in temporal sentence grounding tasks, named Counterfactually Augmented Event Matching (CAEM). This framework enhances the model's counterfactual thinking by generating counterfactual video-text pairs through temporal delays of events in untrimmed videos. Additionally, an Event-Query Matching model is designed to learn joint representations and predict matching scores for each event candidate. The experimental results demonstrate that this method surpasses existing state-of-the-art methods on the Charades-CD and ActivityNet-CD datasets.

**Strengths:**

1. The CAEM method introduced in the paper is innovative, incorporating the concept of counterfactual data augmentation into the temporal sentence grounding task, which is a novel attempt in this field.

2. Extensive experiments are conducted on two widely used datasets, and comparisons are made with several state-of-the-art methods.

3. The paper is well-structured, with clear methodology, detailed experimental design, and thorough analysis of results, making it accessible for readers to understand and replicate.

4. Addressing bias in video understanding has significant potential value for practical multimedia applications.

**Limitations:**

1. The paper could benefit from a deeper theoretical analysis of why counterfactual augmentation helps in this specific task. Including a theoretical justification or a model that explains the observed improvements could strengthen the paper's contributions.

2. The proposed method involves generating counterfactual data, which could potentially increase the computational overhead. The paper does not discuss the scalability of the method or its performance implications on larger datasets. Add a discussion about the scalability of the proposed method, possibly supplemented with experiments on larger or more complex datasets. Evaluating the method's performance in terms of computational resources and time would be valuable for practical applications.

3. To enhance clarity, the paper should detail the relationship between the causal graph and the proposed model. For instance:

    **Define variables clearly** within the causal graph, such as video events, text queries, and their interactions.

    **Map model components** to elements in the causal graph, explaining how each part of the model corresponds to and enforces the causal relationships.

    **Demonstrate causal reasoning** through experiments showing how the model leverages the graph to improve accuracy and bias mitigation.

    **Visualize the causal graph** to aid understanding and provide interpretative insights into the model’s predictions.

**Suitability:**

3

---

### Meta-Review · Area_Chair_XSzz · 2024-06-28

**Recommendation:** Accept (Poster)
**Confidence:** 3

**Metareview:**

This paper is well-written and easy to follow. The task is to localize events in untrimmed videos which is very challenging. The paper proposes a new method called Counterfactually-Augmented Event Matching (CAEM) to solve the training bias problem in the dataset. Its validation is mainly on biased datasets -- which the AC thinks is reasonable. For this exact point, the reviewer ub5m has a different view and his/her final review rating is rejection. The rebuttal has resolved his/her questions (from the AC's understanding), but seems not very convincing to him/her, while he/she did not reply specifically why not convincing.

Given the high quality and novelty of this paper, the AC would like to accept it for publication. Authors are strongly suggested to make revisions according to the reviews for the camera-ready.